

# 1 Predicting tile drainage discharge using machine learning

# 2 algorithms

Saghar Khodadad Motarjemi[1], Anders Bjørn Møller[1], Finn Plauborg[1] and Bo Vangsø Iversen[1]
[1]Department of Agroecology, Aarhus University, Blichers Alle 20, 8830 Tjele, Denmark
*Correspondence to*: Saghar K. Motarjemi (sa.m@agro.au.dk)
**Abstract**
Drainage systems can significantly improve the water management in agricultural fields. However, they may transport
contaminants originating from fertilizers and pesticides and threaten ecosystems. Determining the quantity of drainage
water is an important factor for constructed wetlands and other drainage mitigation techniques. This study was carried out
in Denmark where tile drainage systems are implemented in more than half of the agricultural fields. The first aim of the
study was to predict the annual discharge of tile drainage systems using machine-learning methods, which have been highly
popular in recent years. The second objective was to assess the importance of the parameters and their impact on the
predictions. Data from 53 drainage stations distributed in different regions of Denmark were collected and used for the
analysis. The covariates contained 35 parameters including the calculated percolation and geographic variables such as
drainage probability, clay content in different depth intervals, and elevation, all extracted from existing national maps.
Random Forest and Cubist were selected as predictive models. Both models were trained on the dataset and used to predict
yearly drainage discharge. Results highlighted the importance of the cross-validation methods and indicated that both
Random Forest and Cubist can perform as predictive models with a low complexity and good correlation between predicted
and observed discharge. Covariate importance analysis showed that among all of the used predictors, the percolation and
elevation have the largest effect on the prediction of tile drainage discharge. This work opens up for a better understanding
of the dynamics of tile drainage discharge and proves that machine-learning techniques can perform as predictive models
in this specific concept. The developed models can be used in regard to a national mapping of expected tile drain discharge.
**Keywords:** Tile drainage discharge, Random Forest, Cubist, Cross-validation
**1. Introduction**
Artificial subsurface drainage has a huge impact on the hydrology, nutrient cycling, and sediment dynamics in
agricultural systems (Blann et al. 2009). In temperate climates with fine-textured soils as well as semi-arid regions with



irrigated fields (Ayars et al. 2006), tile drainage is a crucial water management system to control runoff, prevent
waterlogging, and to increase water use efficiency. On the other hand, tile drainage affects both the quantity and the
quality of water resources (Schilling et al. 2012). Nutrient losses and chemical remnants can either be transported
through drains to surface water bodies such as lakes and rivers (Stenberg et al. 2012) or be leached to the groundwater,
and this fresh-water contamination can threaten both human and ecosystem health (Kuzmanovski et al. 2015).
Constructed wetlands are a means to eliminate excessive amounts of nitrogen from drainage water benefiting from
natural nitrate reducing processes in a controlled environment (Messer et al. 2017). These systems are mainly installed to
reduce the pollution from drainage water from agricultural fields and run-off from industrial areas (Magmedov et al.
1996). In order to design constructed wetlands with appropriate sizes, it is necessary to quantify artificial drainage
discharge. Physically-based hydrological models have been developed either to estimate the drainage discharge or to
include it as a component (De Schepper et al. 2017). These models have a common use in academic research and may as
well be used to evaluate various scenarios (Zia et al. 2015). However, they depend on numerous parameters and require
calibration to individual areas (Basha et al. 2008), which makes them complicated and time consuming to apply on a
national scale. Another disadvantage of these models is the conceptualization as the fundament, which leads to invalid
predictions when new empirical data are introduced (Bredehoeft 2005). Beside physically based models, many statistical
approaches have been used to model and to predict state variable such as discharge, but there are limited number of
literature predicting tile drainage discharge with the means of machine learning approaches. This type of data-driven
modelling requires fewer parameters and can perform as an accurate estimation technique and these models have proved
to be flexible and robust enough for many regression applications (Park et al., 2016).
Machine learning is related to computational statistics and is commonly used for predictions based on learning from
historical relationships and trends in the data. Classification and Regression Trees (CART) are a frequently used form of
machine learning models. They work by searching through the covariates of a dataset to find the best splitting single
value. This creates two different groups of data. The process is repeated for the both created groups until a decision tree
forms. Zia et al. (2015) predicted drainage discharge utilizing an M5 decision tree modelling technique on a 17 ha
drained farmland in southern Ireland. Predictions were carried out on a daily basis for a 12-month period. They validated
the suitability of a simplified discharge prediction model for implementation on a system with limited resources.
Kuzmanovski et al. (2015) evaluated machine-learning models in predicting sub-surface tile drainage discharge and
surface runoff on an experimental site in La Jaillière, France using daily data from eleven fields including a reference
field. The dataset was based on meteorological measurements, agricultural practices, and crop management. By



comparing the results from these models with the performance of two physically based models, they found an
improvement in the sub-surface discharge predictions.
In the present study, two different machine-learning models were used to predict yearly tile drainage discharge, Random
Forest (RF) (Breiman, 2001) and Cubist (CB) (Quinlan, 1993). RF is an ensemble approach based on CART (Breiman,
2001). It trains a number of regression trees from bootstrap samples drawn from the original dataset and averages the
results from each tree for the final prediction. The algorithm furthermore introduces randomness into the splitting process
by selecting the optimal split from a random subset of the covariates in each split. CB is a rule-based regression
technique, which does not retrieve one final model like RF but a set of rules related to multivariate models (Walton,
2008). A specific set of covariates will choose an actual prediction model based on the rule that best fits the predictors.
As a commercial and proprietary product, CB has the least algorithmic documentation comparing to random forest.
However, Kuhn et al. (2013) ported it into R, which led to its popularity and it is currently being widely used as a
regression method.
Both RF and CB have been used widely in the recent decades to predict different climatic or environmental parameters.
However, there are few studies, which aim to compare RF and CB models. Walton (2008) estimated urban forest canopy
cover and impervious surface cover using three different models including CB and RF and compared their performances.
They concluded that CB was the best choice for predicting urban impervious surface cover. Noi et al. (2017) compared
the results of Multiple Linear Regression, Cubist Regression, and Random Forest Algorithms in estimation of daily air
surface temperature. They concluded that using different combinations of data, RF or CB algorithms resulted in high
accuracies.
In this study, the chosen methodology is based on machine learning, which is considered as a promising modelling
method in the fields of agriculture and environmental science (Debeljak and Dzeroski 2011). Here we aim to assess the
performance of RF and CB in predicting yearly tile-drainage discharge, to compare the results achieved by both RF and
CB, and to analyze and rank the importance of the covariates.
**2.    Materials and Methods**
**2.1.  Study Area**
Denmark is located in northern Europe with a total area of 42,895 km$^2$, of which 66% are used for agricultural purposes
(Statistics Denmark, n.d.). The climate is temperate with an approximate mean annual precipitation (P) of 770 mm
(Wong, 2013).  The mean temperature is 7.7˚C  ranging from 1.5°C in January to 16.3°C in July. The mean elevation is
31 m above sea level and the landscape is generally flat. The geology divides Denmark into two main areas. An eastern





part with loamy Weichselian moraines and a western part with sandy glacial outwash plains and Saalian moraines.
According to historical maps, wetlands originally covered more than over 20% of the country but due to drainage
activities, they have been reduced in extent during the 19th and 20th centuries.
**2.2. Data**
Data from 53 drainage stations in different locations and regions of Denmark were used in this study (Fig. 1). It included
data from 18 stations established between 2012 and 2016 and historical data from 34 older stations established between
1971 and 2009, of which some are still running and some had been shut down (Hansen & Pedersen 1975; Hansen 1981;
Simmelsgaard 1994; Grant et al. 2009; Kjær et al. 2011; Kjærgaard et al. 2016). Some data originates from ongoing
unpublished drain discharge stations, which have been established in relation to the monitoring of constructed mini-
wetlands. Other data belongs to a former project, iDræn (www.idraen.dk, 2011) where data for some of the stations have
been published earlier (Hansen et al. 2018a,b; Varvaris et al. 2019a,b). For many stations, drainage discharge (Q) was
measured on a daily basis but for some, Q was only measured on a weekly, monthly, or yearly basis. Based on the drain
catchment area, yearly values were converted to a water height per year (mm $y^{-1}$) based on the period from 1 July to the
end of June to incorporate a full hydrological year. Most of the old stations had available data for a range of 19 to 23
years, whereas for some of the new stations there was only data for a few years (1 to 5 years). The lowest discharge (0
mm $y^{-1}$) was recorded in southeast Funen during the year 1995 – 1996, whereas the maximum discharge (1183 mm $y^{-1}$)
was recorded in eastern Jutland during the year 2015 – 2016. The mean discharge for all the stations was 228 mm $y^{-1}$.
The catchment sizes varied from 1 to 164 ha with a mean of 9 ha.

Thirty-seven different covariates were used as predictors (Table 1). Percolation out of the root zone (Db) was calculated
with the simple water balance model EVACROP (Olesen and Heidmann, 1990) driven by input of daily precipitation (P)
and reference evapotranspiration ($ET_0$). This was done since it was not expected that P during the growing season would
contribute to Q due to the high ET during this period minimizing the percolation out of the root zone. However, the
calculated Db is in general closely related to P and Q (Fig. 2). The average Q and the average Db were calculated for
each station to determine the ratio between Q and Db (Fig 3). As shown in Figure 3, for seven stations out of 53, the tile
drainage discharge is more than the percolated water. These stations are located in large catchments often in stream
valleys where external sources (such as regional groundwater) probably flow to the tile drains from outside the
catchment. The absolute amounts of discharged water in all the stations is normalized based on catchment area.





Thirty-three out of 37 covariates were extracted from existing national maps. Topographical variables were calculated by
(Møller et al. 2018) based on a digital elevation model (DEM, Fig. 4A) with a 30.4-meter grid size aggregated from a
DEM with a 1.6-meter resolution. Adhikari et al. (2013) predicted maps of clay contents for the upper two meter of the
soil at a resolution of 30.4 m. These were aggregated by Møller et al. (2018) producing input data in form of maps of clay
content in four depth intervals (Clay A%, Clay B%, Clay C%, Clay D%, Table 1, Fig. 4B). Values of clay content were
also obtained from a national soil profile database using values from the nearest excavated soil profile. Depth to
groundwater (Gwd_model, Table 1, Fig. 4C) was first calculated based on a model at a 500-meter resolution (Henriksen
et al., 2012) and then the groundwater table was resampled to a 30.4-meter resolution using bilinear interpolation (Møller
et al. 2018). Topographic Wetness Index (TWI, Table 1, Fig. 4D) that quantifies topographic controls of basic
hydrological processes (Schillaci et al., 2015) was derived through interactions of fine-scale landform coupled to the up-
gradient contributing land surface area by Møller et al. (2018). A map of soil drainage classes (Møller et al., 2017), a
rasterized choropleth map of geology (Jacobsen et al., 2015), and a map of wetland areas (Wetlands, Table 1, Greve et al.
2014) were also used in the analysis. Horizontal and vertical distances to surface waterbodies were included based on
Møller et al. (2018), who calculated horizontal distances to waterbodies as the two-dimensional Euclidean distance to
vector layers of waterbodies. Hereafter, they calculated the slope to channel as the angle to the hydrologically nearest
waterbody taking into account the surface flow direction. Møller et al. (2018) predicted artificially drained areas
(D_DK_New, Table 1) in Denmark by means of a selective model ensemble including number of geographic variables.
All 37 covariates were used as input to the statistical models.
**2.3.   Models and Measures of Accuracy**
As mentioned earlier, two machine-learning algorithms Cubist (CB) (Quinlan, 1993) and Random Forest (RF) (Breiman,
2001) were used to predict tile drainage discharge.  Cross-validation was used to adjust the parameters of the models and
to assess their predictive accuracy. Cross-validation is a resampling procedure used to evaluate machine-learning models
on a given dataset. For CB, the parameters were adjusted to *committees* and *neighbors*. The parameter *committees* sets
the number of boosting iterations while the parameter *neighbors* set a number of nearby cases, which can be used for
interpolation in order to adjust the predictions. For RF, the parameter *mtry* was adjusted, which sets the number of
randomly selected covariates that are available in each split.
For both algorithms, three different cross-validation procedures were used. Firstly, in order to assess the ability of each
model to predict the tile drain discharge at a new location, *leave-station-out* (LSO) cross-validation was performed. In
this procedure, all the measurements were removed from one station in the data sample and a model was trained from the



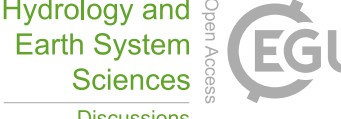

remaining measurements and used to predict Q for the excluded cases. This process was repeated for all stations and
resulting accuracy was calculated.
The stations used in this study are highly clustered in geographic space (Fig. 1). Spatial autocorrelation may therefore
affect the accuracy of the LSO procedure as stations may show similar patterns only because they are located close to
each other. Therefore, a second cross validation procedure as *leave-cluster-out* was used as well, in which the clusters of
stations were left out instead of individual stations. To achieve this, clusters were generated based on the distances
between the stations. Stations located less than 10 km from each other were therefore grouped into clusters. This
procedure resulted in 23 clusters with 1 – 10 stations each. These clusters were later used for cross-validation.
Finally, k-fold cross-validated (KF) RF and CB models were trained on the whole dataset. In this procedure, the dataset
were randomly divided into k disjoint folds, which are approximately equal in size. Each of the folds is used to test the
generated model from the rest of k-1 folds. The performance of the algorithm was evaluated by the average of the
resulting k accuracies from the cross-validation. When a specific value for k was chosen, it could be used in place of k in
the reference to the model, which in this case k = 10 and it could therefore be referred as 10-fold cross-validation (Wong

155 2015).

In total, six models were trained as the CB and RF models were trained separately with *leave-station-out* (LSO), *leave-*
*cluster-out* (LCO), and *k-fold* (KF) cross validations. The accuracy of all five models were assessed with root mean
square error (RMSE):
$$RMSE = \sqrt{\frac{\sum_{i=1}^{n}(Q_{m_i} - Q_{o_i})^2}{n}} \qquad (1)$$
where $Q_{mi}$ is the predicted value of yearly drainage discharge for the *i*-th instance, $Q_{oi}$ is the observed or measured value
of yearly drainage discharge for the *i*-th instance, and n is the total number of instances.
The Nash-Sutcliffe efficiency (NSE) was used for validation as well:
$$NSE = 1 - \frac{\sum_{i=1}^{n}(Q_{m_i} - Q_{o_i})^2}{\sum_{i=1}^{n}(Q_{o_i} - \bar{Q}_o)^2} \qquad (2)$$
where $\bar{Q}_o$ is the mean of observed discharges.
Furthermore, to analyze the effect of the covariates in each model, the covariate importance was extracted from all six
models. The covariate importance measures were scaled to 100% for the most important covariate in each model. In the
beginning, all of the 37 parameters were introduced as covariates to the model. However, the purpose of using machine-





learning is to find a simpler way to predict the target and to determine the most effective parameters, which helps to
reduce the number of covariates and exclude the ineffective ones.

**3.   Results and Discussion**
**3.1.   Model accuracy**
The most accurate predictions were obtained by 10-Fold (KF) cross-validated Cubist (CB) and 10-Fold (KF) cross-
validated random forest (RF) with RMSE of 75 and 77 mm/y and NSE of 0.73 and 0.74, respectively (Fig. 5, Table 2).
According to Singh et al. (2005), an acceptable value for RMSE in hydrological modelling would normally be half of the
standard deviation of training data, which for the current data set was 166 (mm/y). Therefore, leave-station-out (LSO)
cross-validated random forest (RF) with an RMSE of 110 mm/y and LCO cross-validated CB with an RMSE of 113
mm/y could be considered as acceptable models regarding the prediction of Q.
The purpose of performing three different cross-validations was to test the model accuracy with and without the effect of
geological biases. In LSO, a single station containing an entire data set is removed from the training dataset as the target
of prediction. However, the model is still trained on the neighbor stations, which are regionally close to the target. That
could cause overfitting issues. On the other hand, the LCO ensures that on each run of the model, one of the 23 clusters is
excluded as the prediction target, which diminishes the possibility of overfitting caused by geo-regional similarities.
Finally, KF randomly divided the whole dataset into 10 fold with equal size, which does not consider the distribution of
the stations. Data is sampled based on the rows and the difference in size between the training set used in each fold and
the entire dataset is only a single pattern. Each fold contains 41 rows that are selected randomly and each time one of the
10 folds is the validation or test data set. The repeated cross-validation guarantees that different combinations of
randomly selected stations are in different training folds limiting the possibility of overfitting.
With all three cross-validation methods, the accuracies with RF and CB were quite similar. Furthermore, the accuracies
calculated with LSO and LCO are relatively similar, compared to KF, which had a substantially higher NSE and lower
RMSE than the two other cross-validation methods.
**3.2.   Covariate importance**
Results of all the six models indicate that the percolation or discharge out of the root zone (Db) has the largest effect on
the tile-drainage discharge prediction with 100% importance (Fig. 6). The analyses show that elevation (DEM) follows the
Db as the second most important covariate in all the models with more than 80 % importance in LSO-CB and LCO-CB



(Fig. 6 a and b) and between approximately 40 to 50% effectiveness for the other four models (Fig. 6c to f). The clay
content in the D horizon was the third most important covariate in KF-CB and KF-RF (Fig. 6c to f). For the LCO-CB and
LSO-RF models, horizontal distance to the nearest waterbody appears as the third most important covariate with 45% and
21% importance, respectively (Fig. 6a and d). Whereas for the LSO-CB model, clay content in the C horizon and the LCO-
RF model clay content in the B horizon where the third most important covariates (Fig. 6 b and c). The rest of the list
differs between the different models. However, it is observable that for the RF models (Fig 6c to e) only the first covariates
have a significant effect where the rest have less than 20% importance. Nevertheless, for all CB models (Fig 6 a, b, and f)
the top 10 covariate all have more than 20% importance. As previously stated, percolation and elevation have the largest
importance to all of the trained models for the prediction of discharge. Based on the analyses of covariate importance, the
results of the predictions for the two most effective covariates were compared to their measurements (Fig. 7). This
comparison demonstrates how well the models can simulate the relationship between the most important covariates (Db
and elevation) and the prediction target (Q). The open black circles represent the predictors on the x-axis against the
measured drainage discharge (Q) on the y-axis. The red open circles represent the predictors on x axis and predicted
drainage discharge (Q) on the y-axis by each of the six models mentioned on top of the plots. The best match could be
observed on the k-fold cross-validated CB (Fig. 7 e and f).
**3.3. Discussion**
Similar studies targeting the prediction of discharge with machine learning models developed their models in a catchment
scale for time series and chose the daily meteorological data, agricultural practices, and crop management as covariates
(Kuzmanovski et al. 2015, Zia et al. 2015). Also in these studies, they used 10-fold cross-validation to evaluate the
robustness of their model performance. The present study was carried out on a larger scale with catchments of different
sizes distributed in different regions. Along with the percolation, a number of different geological features were used as
input parameters to assess if it is possible to predict the tile drainage discharge based on spatially variable geophysical
characteristics of the different sites. In the few similar studies (Rasouli et al. 2012, Kuzmanovski et al. 2015, Zia et al.
2015), the study area was either one specific catchment or few fields or catchments very close to each other. This means
that the geological features were similar. Being able to train machine-learning models on different catchments in very
different locations had enabled us to make use of differing geographical characteristics as predictor variables. Predictions
were carried out in a yearly basis and were cross-validated with three different methods.
The accuracies of RF and CB models in comparison to each other for all the cross-validation methods were quite similar.
On the other hand, the obtained accuracies from LSO and LCO are relatively similar but lower compared to KF, which





had a substantially higher NSE and lower RMSE than the two other cross-validation methods. The higher accuracies
achieved by KF is most likely results from having the observations of a given station from other years during the
prediction procedure. The accuracy obtained with KF could be considered as the internal accuracy of the model, while
LSO and LCO better represent the accuracies at new locations without previous measurements of tile drain discharge at
the same station. The proposed tile-drainage discharge predictive model is not dependent on the climatic and constantly
measured data and makes it possible to use different geographical properties as predictive parameters.
Logically, Db is the main driving variable since it takes into account water lost by evaporation from the soil surface,
transpiration of water by the crop, and the increase of water stored in the soil. During the growing season, a high value of
P will not necessarily lead to a corresponding high value of Q since it is only the part of P that infiltrate out of the root
zone that potentially can flow into the tile drains. It is also expected that the clay content in the soil, especially the clay
content in the lower horizons below tile drain depths, would have an effect on the drain discharge. A high clay content in
the subsoil would lead to a secondary groundwater table building up outside the growing season to the level of the tile
drains. That the clay content not play a more important role as a covariate might be explained by the relatively high
prediction error of the clay content especially at lower depths for the used soil maps.
The position of the tile-drained field in the landscape will have an effect on the tile drain discharge. At low positions in
the landscape, the flow of water to the drains is expected to be relatively high due to a high contributing area of expected
incoming regional groundwater generated from a larger area outside the tile-drained field. Such areas are also indicated
in Figure 3 corresponding to high values of Q/Db. On the other hand, at higher positions in the landscape with no or only
a minor contribution of regional groundwater, a proportional part of the water infiltrating into the drains is generated
mainly locally from water percolating out of the root zone (Db). It was expected that DEM derived indices such as TWI
or SagaWI (Table 1) would describe more precisely the contribution of water in the tile drains and therefore supposed to
be important covariates. Both indices attempt to describe the hydrological flow paths in the landscape and should be able
to identify areas with a high contribution of water flowing to the drains. However, only for the k-fold cross-validated RF
model (Fig. 6E), TWI is found within the list of the top 10 most import covariates. On the other hand, DEM is placed as
the second most or the most important covariate for all models. This proves that the position in the landscape does have
an effect on the tile drain discharge. That the derived topographical indices only play a minor role in the statistical
models might be related to the fact that it can vary considerably within the individual drained catchments. On the other
hand, other derived DEM indices such as valley depth (Valldepth), vertical distance to the nearest waterbody (Vdtochn),
horizontal distance to the nearest waterbody (Hdtochn), and downhill gradient to the nearest waterbody (Slptochn) are all
found in the top 10 list.



By applying input from a distributed model predicting Db it is possible to apply the developed model on a national scale
developing maps that can be used as a tool to predict the yearly drain discharge. National water resource models in
Denmark exists that can be used for such purposes (e.g. Højberg et al. 2013). Outputs from the model can be based on
averages for a certain period. Also, the possible variation between years as well as outputs in relation to future climatic
scenarios can be studied.

**4.   Conclusion**

For the current study, two different machine-learning models (RF and CB) were applied on a relatively big dataset
containing measured yearly drainage discharge (Q) and 37 parameters as covariates and the results indicated a successful
implementation. The predictive models were trained on 53 drainage stations distributed all over Denmark with different
characteristics and multiple years of data and cross-validated with three different methods. The best results were
achieved by k-fold (KF) cross-validated Cubist (CB) and random forest (RF) and the performance measures certifies the
results. RMSE and NSE of both models indicates a good accuracy of the predictive models based on the hydrological
modelling standards. Instead of physically-based models that acquire numerous parameters, machine learning models
could perform as strong tools for quantifying the tile-drainage discharge with lower complexity. In this study, percolation
or discharge out of the root zone (Db) calculated with the simple water balance model EVACROP, and elevation (DEM)
where the most important covariate for predicting yearly discharge. Finally, it was concluded that considering the
distribution of stations, the method of sampling and the cross-validation has a large effect on estimates of model
accuracies. The developed model can be used in relation to a national mapping of yearly tile drain discharge.

**Acknowledgments**

This study was supported partly by the Innovation Fund Denmark project Future Cropping (www.futurecropping.dk) and
partly by the Ministry of Environment and Food of Denmark GUDP project iDræn (idraen.dk). We are grateful to David
Nagy for his constant support and helpful comments.

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



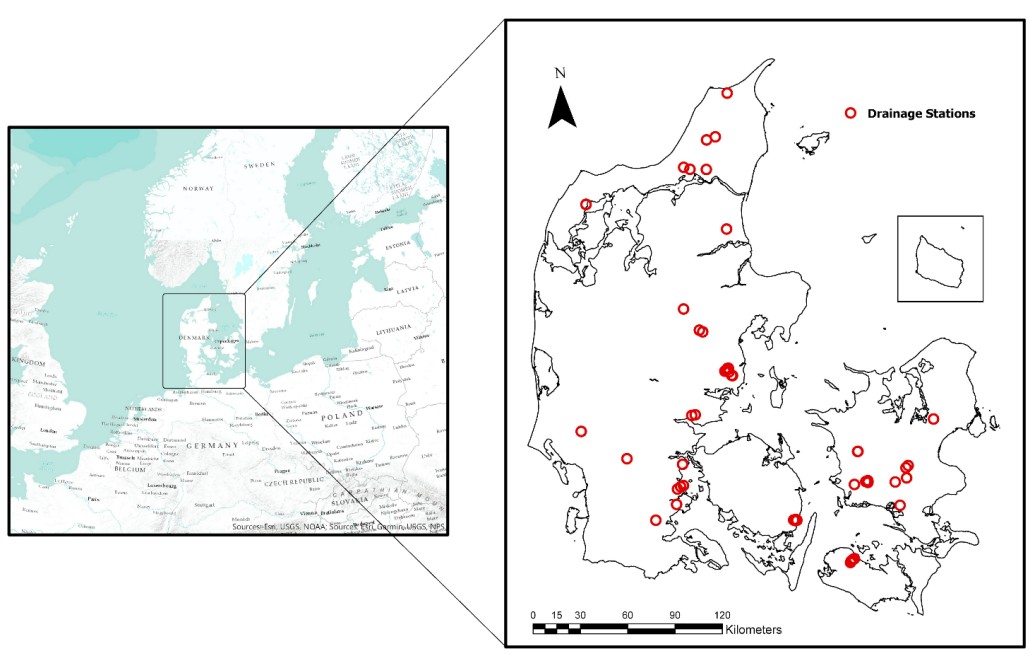


**Figure 1. Study area and the location of the 53 drainage stations throughout Denmark**

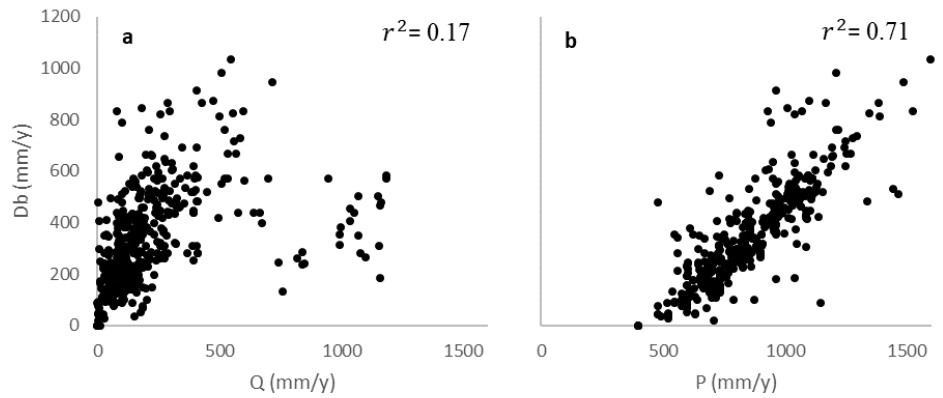


**Figure 2. a) Measured yearly drainage discharge (Q) against calculated percolation (Db) b) Observed precipitation (P) against**
**calculated percolation (Db)**






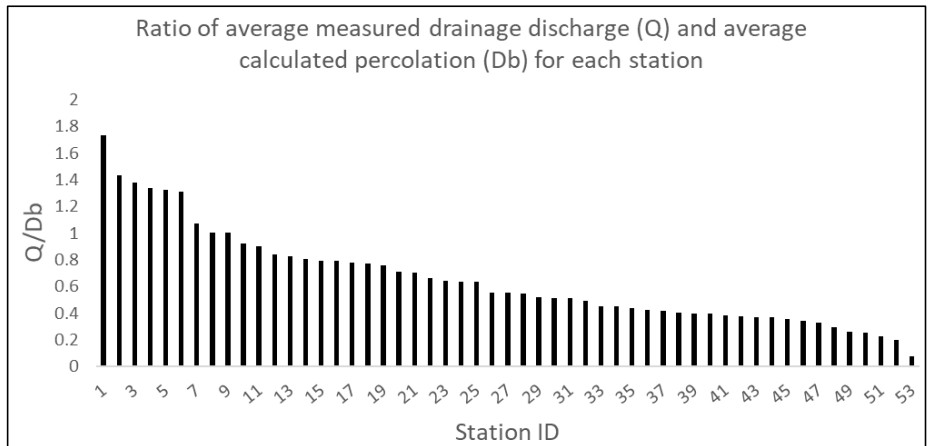


**Figure 3. Ratio of average measured drainage discharge (Q) and average calculated percolation (Db) for each station**

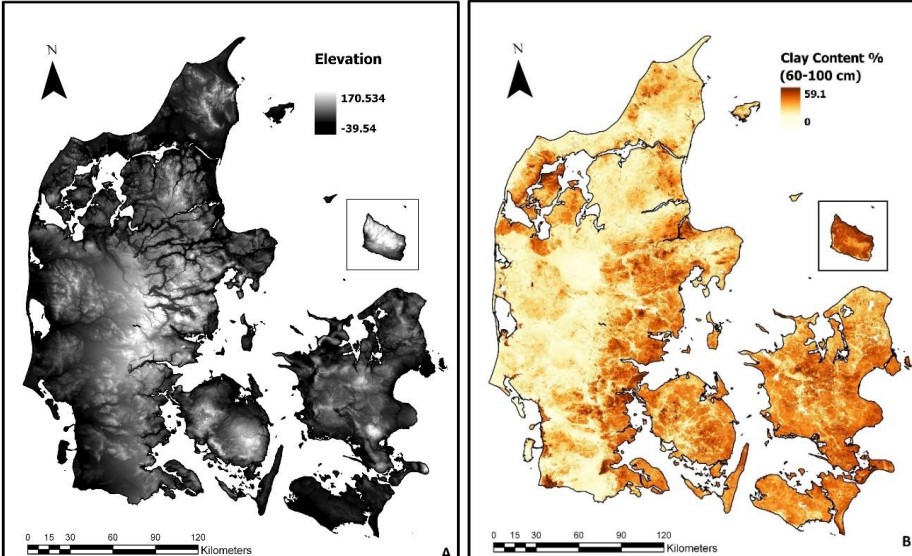


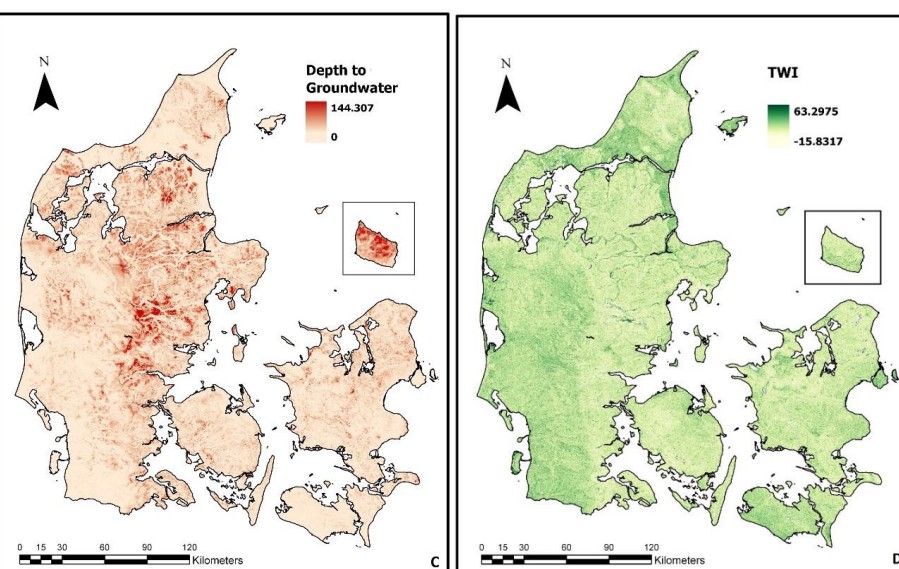


**Figure 4. A. Elevation based on a Digital Elevation Map (DEM).  B. Aggregated clay content in the C-horizon (Møller et al.,**
**2018) C. Interpolated depth to groundwater (Møller et al., 2018) D. Topographical wetness index (Møller et al., 2018)**



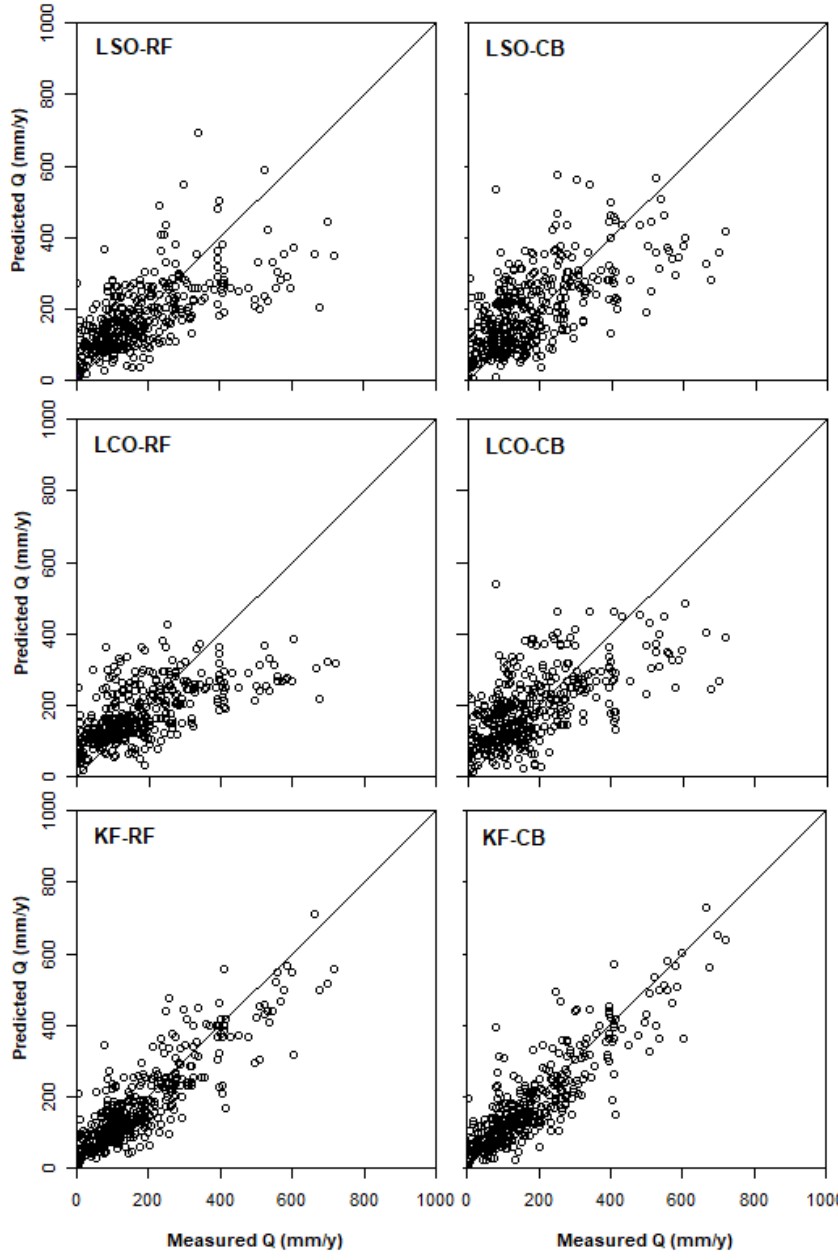


**Figure 5. LSO-RF: Leave station out cross-validated random forest model. LSO-CB: Leave station out cross-validated cubist**


**model. LCO-RF: Leave cluster out cross-validated random forest model. LCO-RF: Leave cluster out cross-validated cubist**


**model. KF-RF: K-Fold cross-validated random forest model. KF-CB: k-fold cross-validated Cubist model.**




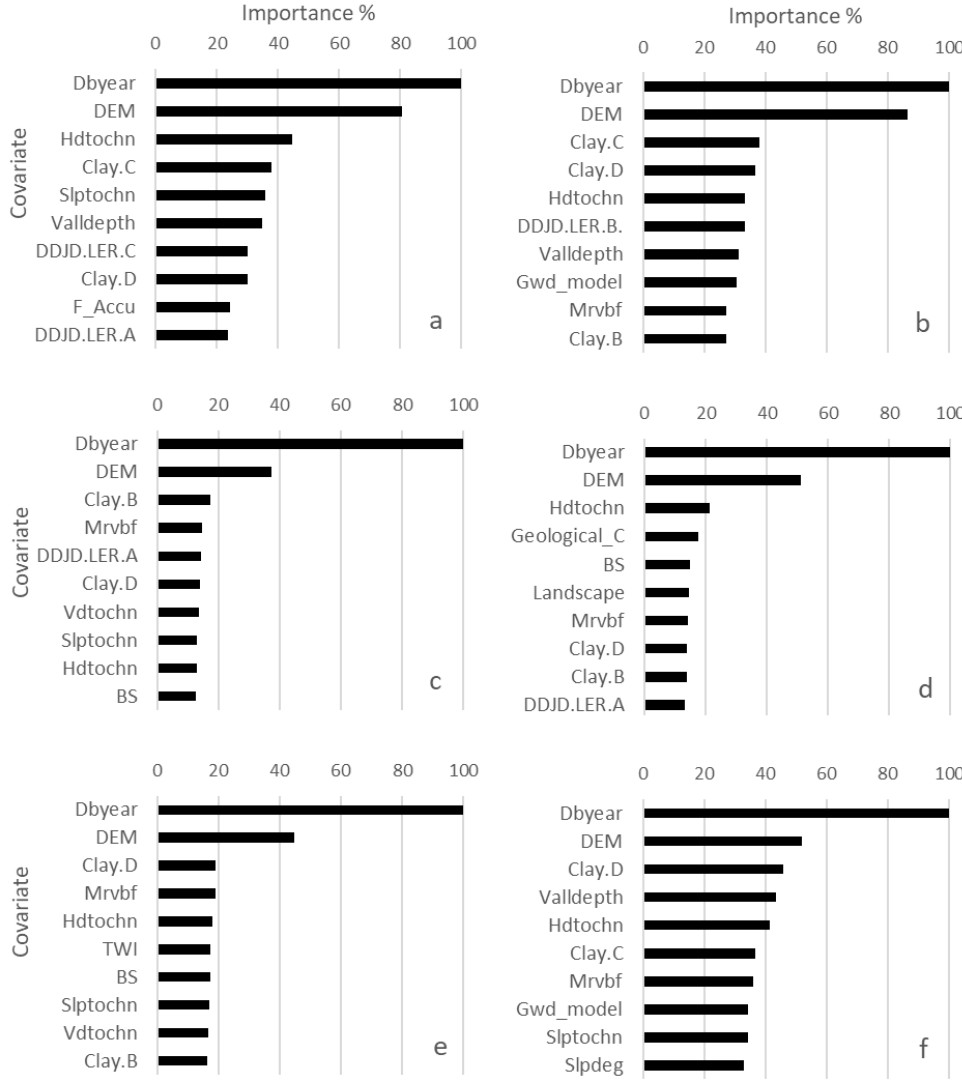


**Figure 6. a) Top 10 most important covariates of the leave-cluster-out cross-validated CB model b) Top 10 most important covariates of the leave-station-out cross-validated CB model c) Top 10 most important covariates of the leave-cluster-out cross-validated RF model d) Top 10 most important covariates of the leave-station-out cross-validated RF model E) Top 10 most important covariates of k-fold cross-validated RF model F) Top 10 most important covariates of the k-fold cross-validated CB model.**




**Figure 7. a, c, and e) Measured discharge against calculated percolation in black open circles, predicted discharge against**


**calculated percolation in red open circles for the selected models with the best prediction. b, d, and f) Measured discharge against**






elevation in black open circles, predicted discharge against elevation in red open circles for selected models with the best
prediction
**Table 1. List of covariates used to predict the discharge including a description of the parameter and a range specifying the**
**type of covariate.**

| Predictors | Description | Range/ Class |
|---|---|---|
| Db | Percolation/Discharge out of the root zone (mm y⁻¹) | 0 – 1033 |
| Geological_R | Geological region | 7 classes |
| DEM | Elevation (m) | 0.74 – 83.16 |
| Geological_C | Geology of the area | 10 classes |
| F_Accu | Flow Accumulation/Number of unslope cells | 1 – 1108 |
| SagaWI | SAGA Wetness Index | 12.16 - 16.58 |
| TWI | Topographic Wetness Index | 3.47 – 12.33 |
| BS | Depth of Sink (m) | 0 – 2.17 |
| D_Class | Drainage class | 5 classes |
| Clay A %† | Clay content 0-30 cm soil depth | 3 – 20.3 |
| Clay B %† | Clay content 30-60 cm soil depth | 2 – 29.1 |
| Clay C %† | Clay content 60-100 cm soil depth | 1.5 – 31 |
| Clay D %† | Clay content 100-200 cm soil depth | 2.2 – 32.6 |
| DDJD LER-A%‡ | Clay content in A horizon | 3 – 24.8 |
| DDJD LER-B%‡ | Clay content in B horizon | 0 – 31.97 |
| DDJD LER-C%‡ | Clay content in C horizon | 0 – 29.1 |
| JB | Danish soil classification for the A horizon | 12 classes |
| Gwd_Int | Depth to groundwater table interpolated from well observations and surface water (m) | 0 – 25.31 |
| Wetlands | 0: Non-wetlands; 1: Wetlands; 2: Central wetlands; 3: Peatlands. | 4 classes |
| D_DK_New | Artifical drainage-new map | 2 classes |
| DP_New | Drainage probability-new map | 0 – 0.86 |

| D_DK | Artifcial drainage-old map | 2 classes |
|------|----------------------------|-----------|
| DP | Drainage probability-old map | 0 – 0.82 |
| Demdetrend | Elevation minus the mean elevation in a 4 km radius (m) | -11.4 – 26.04 |
| Dirinsola | Direct insolation (kWh/year) | 1150.08 – 1348.61 |
| Gwd_model | Depth to groundwater from the model (m) | 0 – 32.42 |
| Hdtochn | Horizontal distance to the nearest waterbody (m) | 0 – 1114.89 |
| Midslppos | Mid-slope position | 0 – 0.7 |
| Mrvbf | Multi-resolution index of valley bottom flatness | 0.07 – 8.68 |
| Slpdeg | Surface slope gradient (degrees) | 0.09 – 7.53 |
| Slptochn | Downhill gradient to the nearest waterbody (m) | 0 – 3.48 |
| Vdtochn | Vertical distance to the nearest waterbody (m) | 0 – 19.28 |
| Valldepth | Valley depth (m) | 2.43 – 21.35 |
| Landscape | Landform types | 11 classes |

† From the map of Adhikari et al. (2013); ‡from the national soil database
**Table 2. Error summary of six trained models**

| Error \ Model | LSO-CB | LCO-CB | LSO-RF | LCO-RF | KF-RF | KF-CB |
|---------------|--------|--------|--------|--------|-------|-------|
| RMSE | 116.53 | 115.04 | 110.65 | 115.82 | 76.05 | 70.98 |
| NSE | 0.37 | 0.39 | 0.44 | 0.38 | 0.73 | 0.74 |

