# Peer review of "Predicting tile drainage discharge using machine learning"

_Hydrology and Earth System Sciences, 2019_

## Referee Comment (RC1) · Anonymous Referee #1 · 23 Jan 2020

The manuscript entitled "Predicting tile drainage discharge using machine learning algorithms" is a well-written manuscript, the objectives are clearly stated, the Introduction section provides a sufficient description of the machine learning algorithms used in this study also to non-experts, and paragraphs are concise and logically connected. Although I appreciated the relatively short length of the manuscript, the methods need to be expanded and clarified. This research reports the application of machine learning to predict yearly discharge values in tile drains in Denmark. The Authors integrated a sufficient number of datasets from multiple sources to build the predictive model. However, the number of measurements reduces to 53 values and the datasets appear to be redundant rather than distinct. As a consequence, my concern is that the model may not be able to capture interaction effects and nonlinear dependencies as well as it

could not really learn more information from similar datasets and possibly made poor decisions in ranking the datasets.

I believe the research would benefit from a deeper analysis of the effects of using stations with different ranges of historical measurements and the significance of the datasets, which may altogether bias the model. Figures and Tables might be revised to increase the clarity, the level of detail, and to ease Readers' understanding of the possible research limitations. The manuscript can be interesting for the scientific community working on machine learning applied in hydrology. But at the present state, I would not recommend it for publication because it is not possible to understand the soundness of the methods used to train the model. I am willing to review a revised version of the manuscript.

Below I report my major concerns of the methodology.

Abstract

L20: I have to disagree that this work opens up for a better understanding of the dynamics of the drainage discharge. As I discuss later, the Authors use average values throughout the observation period, and therefore, dynamic effects are loss.

Section 2.2

The Authors report They used data from 53 stations; 18 stations collect data from 2012 to 2016 and 34 between 1971 and 2009. The number of stations does not sum up to 53. The problems are that the measurements cover different time periods. The model may have been trained using data from 2016 in one location and data from 1971 in another location. There are no information on discharge trends in the stations, which may be available at those locations with long observations. Anyway, yearly values were calculated for each location neglecting possible trends and variance in data measurements. Can the Authors find a methodology to use only stations that are consistent in time? There are no information on data quality, while possible data gaps may exist. If

this was the case, how were they filled?

May I please ask the Authors to add a reference to indicate where precipitation measurements and evapotranspiration values come from?

According to the Authors' hydrological model, percolation out of the root zone is calculated as the difference between precipitation and evapotranspiration. Here, there exist some assumptions which have not been stated. For example, is it valid to neglect irrigation from the model? Is it valid to assume that the crop-specific coefficient $K_c=1$ to calculate the actual evapotranspiration from the potential one?

L113: May I please ask the Authors to report the accuracies of the digital maps in Table 1? In this regard, the Authors comment at L237 that accuracy error of the digital maps may influence their importance as covariates. If the Authors know the accuracies, They could carry out a sensitivity analysis using the available standard deviation as prior information and assess the prediction outcomes.

Section 2.3

How did the Authors integrate numerical and categorical variables? What was the approach followed by the Authors to convert categorical variables to numerical ones? May the Authors discuss what are the implications of such integrations with respect to the final predictions?

L156: Can I please ask the Authors to state how They extracted the covariate importance? In general, which software and packages were used to carry out the study?

L169: The Authors report the possibility to use methods to determine the most effective parameters, thus opening the opportunity to reduce the number of covariates. Did the Authors try to rerun the machine learning using a subset of covariates?

Section 3.1

Please add a reference to Table 2 where the accuracies of the methods are reported.

L179: The cluster analysis was an interesting approach. However, the clusters were different in size. Was there a relationship between overall accuracy and number/location of the stations excluded from the training set?

Section 3.2

L193: Can I please ask the Authors to use one term either percolation or discharge out of the root zone, for clarity?

L211: Please use a new Section for the Discussion.

L225-L230: This paragraph seems crucial for the understanding of the predictions but it is difficult to follow. The Authors here discuss the implications of having time-series covering different time-periods. Because Their explanation is not clear, it is difficult to be convinced about Their interpretation.

L229: The Authors state that the model is not dependent on climatic forcing. However, this is not because the effects of precipitation and evapotranspiration are accounted for, which are climatic variables.

L241: How can the Readers know that the areas with high catchment area are the ones with larger Q/Db? The Authors may use Figure 3 to show such relation? Maybe They could add some text to report the area.

L248: While it is possible that low-elevated areas are the ones with higher Q, it is difficult to think that distance from groundwater table or the depth to sink (does this refer to the depth to tile drain?) are not significant. Have the Authors tried to remove the DEM as covariate and see how the other covariates rank?

L258: I have to disagree with the Reviewers statement: "the possible variations between years as well as outputs in relation to future climatic scenarios can be studied". In fact, there is no analysis with regards of time; yet, the Authors are somehow contradicting Themselves as They stated at L229 that the model does not depend on climatic forcing. Therefore, no studies in relation to future climate scenarios can be carried out.

As far as the Figures are concerned:

Figure 1

It can be more informative. It would be ideal to have an ID that identifies each location and link the spatial information with the corresponding metadata compiled in a large additional table.

It shows that areas are quite far from each other and may follow peculiar dynamics. This strengthen my concern that the number of yearly values and covariates may not suffice to highlight the functioning of the system. Showing how the location cluster affect the predictions may support the interpretation of the results.

Figure 2

I would suggest to first show the relationship between measured variables (i.e., Q as dependent variable and P and ET0 as independent). At a later stage I would show the relationship between measured Q and predicted Q. Finally, I would show the relationship between Q and significant covariates. As of now, Figure 2b is not informative. It shows the relationship between the Discharge out of the root zone (Db) and the Precipitation, which the latter was used to calculate Db.

Figure 3

It would be more meaningful if additional information were provided to understand for which conditions Q is greater than P (e.g., low-lying areas, etc. . .).

Figure 4

I like the idea of showing the maps because they report the gradient The Authors could ease the Readers if the locations were indicated in the maps. It might be valuable to create insets and show scatterplots between measured Q with each covariate reported in the map, with an errorbar to indicate the accuracy of the maps at the location.

Figure 5

This figure makes me wonder: why if I do not use 1 station in the training set (LSO), I have worse accuracy than when I do not use 10% of the stations in the training set (KF). I think the Authors very briefly discussed this at L225. But I would kindly ask Them to further clarify and expand their analysis on this. Was it about the time coverage? The location? The accuracy of the maps? I believe it is possible to disentangle this.

Figure 7

Panels b,d,f show elevation, which seems to have big range-discrete values. Can the Authors please explain?

Table 2. Is there a p-value? Such value could be used to decide which features are significantly relevant and control possible false discovery rates.

MINOR COMMENTS

L57: Can I please ask the Authors to state by how much was the improvement in terms of performance of the machine learning compared to physically based models to predict tile drainage discharge?

L73: Can the Authors please summarize the accuracy to the Readers?

L156: Note that, the Authors are not using 6 models, but 2 models and validating each with 3 different methods.

---

## Referee Comment (RC2) · Anonymous Referee #2 · 6 Feb 2020

The manuscript describe a study about modelling tile-drainage discharge aggregated on annual level on different sites in Denmark, by using machine learning algorithms, in particular Random Forest and Cubist. The overall presentation of the study is fairly concise. However, the manuscript lacks more detailed explanation on motivation of such study and final conclusions on applicability of the results. The latter is most probably the case due to the miss-conception of the validation process. More details on the study and manuscript's sections are given below.

The study itself has been thought systematically on how to approach the modelling phase. However, some phases were misconducted. First of all, the time scale of the study is considered to be annual in regard with the output, which is not clearly specified how then the input has been encoded/aggregated, knowing the fact that meteo data

[Figure]

are available on daily bases. Next problematic approach is using mechanistic models to encode/represent the input in the modelling process. Such case is with meteorological data that are run through water balanced model EVACROP. Finally, after performing the cross validation, the study does not extract any new knowledge, rather discuss differences in cross-validation techniques - which clearly does not fit the scope of the journal. To this end, I would rather say that finding out that percolation (Db) is most important attribute upon running huge machinery is not an added value, as that fact is proven by theory and more specific by correlation of both variables discharge and percolation, which is obvious from Figure 7. Rather more interesting contribution would be to see which of precipitation and/or evapotranspiration is more significant in combination with different/specific landscape and soil characteristics. Similarly, second most importantly identified covariate - elevation - is pretty difficult to be simply explained as cause for discharge. The small range of values with pretty small sample size cause a behaviour as a clustering bias, especially if experimental sites are uniformly (equidistantly) distributed along the given range. So instead of discovering more interesting patterns, those are replaces with single covariate that encapsulate different processes under the hood. Therefore, I would rather see what is happening if this covariate is removed.

Regarding the manuscript, the sections introduction and data are well described. The methodology and validation part is also fairly good described, except the part for how the importance of covariates is performed - especially knowing the fact that RF is not that open model so to be able to easily extract the most important covariates. Results and discussion section are lacking more details and focus on actual findings and less (or at least not that dominantly) on performance from different validation schemas. Validation schemas are well defined, and in discussion difference in performance of the models should be discussed - talking of which, spatial bias is not mention upon introduction. Such given discussion sounds more of evaluating three different validation schemas, rather than discussion of new findings in the domain of hydrology.

[Figure]

Finally, few sentences need to be strongly changed as they are not true:

l.20: "This work opens up for a better understanding of the dynamics of tile drainage discharge and proves that machine-learning techniques can perform as predictive models in this specific concept." - too optimistic conclusion without good ground for such claim.

l.229: "The proposed tile-drainage discharge predictive model is not dependent on the climatic and constantly measured data and makes it possible to use different geographical properties as predictive parameters." - this is absolutely not true as percolation is derived from a model that uses at input precipitation and evapotranspiration data.

---

## Editor Comment (EC1) · Christian Stamm (Editor) · 10 Feb 2020

Editor comment

HESSD-Manuscript **"Predicting tile drainage discharge using machine learning algorithms"** (HESS 2019-650).

Dear Dr. S. Motarjemi

I'd like to add a few further remarks in addition to what the other two reviews commented.

[Figure]

**L. 83:** The year of the citation of Wong is inconsistent between the main text and the reference list.

**L. 156 - 157** The number of models seems to be inconsistent.

**L. 190 - 191:** What could be reasons why KF performs worse? Is there a spatial bias?

**L. 238:** What is the basis for this statement?

**L. 244:** Does the predictive value of DEM-derived indices not depend very much on the spatial support and resolution of the data? Have you calculated these indices as averages across the catchments?

**Fig. 1a:** The data seem to separate into two clusters. Do the points with high discharge but rather low percolation have something in common that could explain the differences?

**Fig. 1a&b:** Combining the two data suggests that drainage discharge is well correlated (and predicted) by the amount of precipitation. How does this relationship look like if you additionally distinguish between clay and sandy soils?

**Tab. 1:** Please provide the distribution of predictors (as supporting information).

Sincerely

Christian Stamm

---

## Author Comment (AC1) · 21 Feb 2020

We would like to thank the referee for all the useful and constructive comments, which would enable us to improve the manuscript. Precision and the detailed assessment is very much appreciated. In the following, we provide replies to the referee's comments. We also describe the changes, which we will make in the final manuscript to accommodate the referee's comments.

1: Referee comment: However, the manuscript lacks more detailed explanation on motivation of such study and final conclusions on applicability of the results. The latter is most probably the case due to the miss-conception of the validation process. Finally, after performing the cross validation, the study does not extract any new knowledge,

rather discuss differences in cross-validation techniques - which clearly does not fit the scope of the journal.

Authors' reply: We agree that such explanation is missing in the current version of the manuscript and it would be included in the revision. The main objective of this project is to produce a map of yearly tile drainage discharge on a national scale. More than half of the agricultural fields in Denmark are artificially drained, and studying the pesticide and nutrient leaching through the tile drains has been a focus for many years. Such studies requires costly measurements of drainage discharge at different scales and a discharge map could be used as a fair proxy. On the other hand, mitigation techniques such as constructed wetlands have been widely used in the recent decade targeting the removal of excessive nutrients from the drainage water, before it reaches the fresh water reservoirs. Quantification of drainage discharge is of high importance for the design of the constructed wetlands and a national scale map could assist in this regard. The current study employed all the available data from the stations representing different geological regions with various characteristics to train models that can perform as predictive tools, applicable to the new datasets. The detailed analysis on the validation process is to stress the importance of resampling procedure and the spatial bias that it could cause in this type of data. Most of the 53 stations have multiple years of measured drainage discharge and unlike percolation, the other covariates are constant with time. Training the model on 90 % of the data increases the possibility of having the same station in the training dataset and in the test dataset. Leave-station-out guarantees that the target station does not also appear in the training dataset, however, it would still bias the accuracy assessment as it has similarities with neighboring stations. Which is why leave-cluster-out resampling is the least biased when training the model, as it excludes all the stations within 10 km (as a cluster) from the training dataset. Finally, the importance of resampling methods in the application of machine learning in hydrology has not been widely discussed (to our knowledge) and the findings of this study could be of relevance.

2: Referee comment: The study itself has been thought systematically on how to approach the modelling phase. However, some phases were misconducted. First of all, the time scale of the study is considered to be annual in regard with the output, which is not clearly specified how then the input has been encoded/aggregated, knowing the fact that meteo data are available on daily bases. Next problematic approach is using mechanistic models to encode/represent the input in the modelling process. Such case is with meteorological data that are run through water balanced model EVACROP.

Authors' reply: The percolation (Db) was calculated on a daily basis based on the precipitation and evapotranspiration (for winter wheat as the most dominantly cultivated crop in Denmark), and later was summed up for the entire hydrological year as a yearly value. The drainage discharge data was mostly available on a yearly basis and for those on a daily scale; same method was used to calculate the yearly values. The aforementioned information will be included in the revised manuscript. The importance of the Db calculations via the simple water balance model EVACROP will be discussed during further replies.

3: Referee comment: To this end, I would rather say that finding out that percolation (Db) is most important attribute upon running huge machinery is not an added value, as that fact is proven by theory and more specific by correlation of both variables discharge and percolation, which is obvious from Figure 7. Rather more interesting contribution would be to see which of precipitation and/or evapotranspiration is more significant in combination with different/specific landscape and soil characteristics. Similarly, second most importantly identified covariate - elevation - is pretty difficult to be simply explained as cause for discharge. The small range of values with pretty small sample size cause a behaviour as a clustering bias, especially if experimental sites are uniformly (equidistantly) distributed along the given range. So instead of discovering more interesting patterns, those are replaces with single covariate that encapsulate different processes under the hood. Therefore, I would rather see what is happening if this covariate is removed.

Authors' reply: The most important reason for using calculated percolation instead of measured precipitation is the summer precipitation events, which would not affect the drainage discharge. Denmark has a precipitation deficit in the summer, and the root zone therefore stores most of the precipitation during the summer. Precipitation is distributed over the entire year. Large precipitation events occur both during the winter half year as well as during the summer half year. In our study, we aim at predicting the yearly drain discharge due to the low temporal resolution of a large portion of our drainage discharge data set. Using the EVACROP-simulated Db instead of precipitation will eliminate drain discharge during the growing season where large precipitation events seldom will trig any significant tile drain discharge events. However, the model was also ran with the precipitation as a covariate instead of percolation and the results of model accuracies and the most important covariates are presented in Figures 1 and 2, respectively. The accuracy of the models are not significantly different compared to the results reported on the manuscript with Db as covariate. In Figure 7 of the first version of the manuscript, tile drainage discharge shows a U-shaped relationship with elevation. This shows that topography can have an effect on discharge, which Db does not account for. As we used the 1-dimensional EVACROP model to calculate Db, this covariate does not account for topography. The observed pattern is most likely a combination of several effects. Firstly, higher elevations receive more precipitation, which would increase discharge. Secondly, EVACROP does not account for surface flow. Lower elevations are likely to receive additional water from upslope positions, which would increase discharge. Thirdly, lower elevations will often have a shallower depth to the groundwater, and groundwater flow from higher positions may therefore contribute to the increased discharge. Together, these explanations show why intermediate elevations may have less discharge than higher and lower elevations. To assess the effect of excluding DEM, the model was run without the covariate and the results are shown in Figures 3 and 4. The accuracy of the models do not show a noticeable change after excluding the DEM as a covariate. Regarding the most important predictors, horizontal distance to the channel (Hdtochn) and clay content in the D horizon (Clay.D) appear as

the second and third most important variables after precipitation. These two covariates also had high importance in the models that used elevation as a covariate. The results mainly show the adaptive behavior of machine learning models. When an important covariate is missing, the algorithms can to some extent use other correlated covariates to act as proxies. For example, valley depth and vertical distance to channel may act as proxies for elevation. Presented results in this discussion could be also included in the paper to provide a comparison.

4: Referee comment: Regarding the manuscript, the sections introduction and data are well described. The methodology and validation part is also fairly good described, except the part for how the importance of covariates is performed - especially knowing the fact that RF is not that open model so to be able to easily extract the most important covariates.

Authors' reply: We fully agree that important information about the importance measures of the predictors is missing on the manuscript and will be included after revision. We chose %IncMSE as the measure of variable importance in the RF model. The %IncMSE indicates the increase in the MSE of prediction, drainage discharge in this study, as a result of one variable being permuted. The higher the value of %IncMSE is, the more important the variable is for the regression of the RF model. For the Cubist model, each predictor had a value of the VarImp (%), which is a linear combination of the usage of each variable in the rule conditions and the linear regression models. We used this value to measure the importance of each predictor in the Cubist model. We calculated these two measures using the function varImp in R package caret, which we used for training the models.

5: Referee comment: Results and discussion section are lacking more details and focus on actual findings and less (or at least not that dominantly) on performance from different validation schemas. Validation schemas are well defined, and in discussion difference in performance of the models should be discussed - talking of which, spatial bias is not mention upon introduction. Such given discussion sounds more of evaluating three different validation schemas, rather than discussion of new findings in the domain of hydrology.

Authors' reply: The recommendation will be taken into account for the revision of the manuscript and necessary changes mentioned by the referee will be implemented.

6: Referee comment: l.20: "This work opens up for a better understanding of the dynamics of tile drainage discharge and proves that machine-learning techniques can perform as predictive models in this specific concept." - too optimistic conclusion without good ground for such claim.

Authors' reply: We fully agree that the sentence is formulated in a wrong way. As the discharge is annual in different time periods for each station, the study cannot open up for better understanding of the "dynamics".

7: Referee comment: l.229: "The proposed tile-drainage discharge predictive model is not dependent on the climatic and constantly measured data and makes it possible to use different geographical properties as predictive parameters." - this is absolutely not true as percolation is derived from a model that uses at input precipitation and evapotranspiration data.

Authors' reply: We fully agree with the comment and the statement will be excluded from the manuscript.

Please also note the supplement to this comment:
https://www.hydrol-earth-syst-sci-discuss.net/hess-2019-650/hess-2019-650-AC1-supplement.pdf

———————————————

[Figure]

**Fig. 1.** Model performance of KF-RF: K-Fold cross-validated random forest model, KF-CB: k-fold cross-validated cubist model with precipitation (P) instead of percolation (Db).

[Figure]

**Fig. 2.** Model performance of LSO-RF: Leave station out cross-validated random forest model, LSO-CB: Leave station out cross-validated cubist model with precipitation (P) instead of percolation (Db).

[Figure]

**Fig. 3.** Model performance of LCO-RF: Leave cluster out cross-validated random forest model, LCO-CB: Leave cluster out cross-validated cubist, with precipitation (P) instead of percolation (Db).

[Figure]

**Fig. 4.** Top 10 most important covariates of KF-RF: K-Fold cross-validated random forest model, KF-CB: k-fold cross-validated cubist model, with precipitation (P) instead of percolation (Db).

[Figure]

**Fig. 5.** Top 10 most important covariates of LSO-RF: Leave station out cross-validated random forest model, LSO-CB: Leave station out cross-validated cubist, with precipitation (P) instead of percolation (Db).

[Figure]

**Fig. 6.** Top 10 most important covariates of LCO-RF: Leave cluster out cross-validated random forest model, LCO-CB: Leave cluster out cross-validated cubist, with precipitation (P) instead of percolation (Db).

[Figure]

**Fig. 7.** Model performance of KF-RF: K-Fold cross-validated random forest model, KF-CB: k-fold cross-validated cubist model, when models was ran after excluding DEM.

[Figure]

**Fig. 8.** Model performance of LSO-RF: Leave station out cross-validated random forest model, LSO-CB: Leave station out cross-validated cubist model, when models was ran after excluding DEM.

[Figure]

**Fig. 9.** Model performance of LCO-RF: Leave cluster out cross-validated random forest model, LCO-CB: Leave cluster out cross-validated cubist, when models was ran after excluding DEM.

[Figure]

**Fig. 10.** Top 10 most important covariates of KF-RF: K-Fold cross-validated random forest model, KF-CB: k-fold cross-validated cubist model, when models was ran after excluding DEM.

[Figure]

**Fig. 11.** Top 10 most important covariates of LSO-RF: Leave station out cross-validated random forest model, LSO-CB: Leave station out cross-validated cubist, when models was ran after excluding DEM.

[Figure]

**Fig. 12.** Top 10 most important covariates of LCO-RF: Leave cluster out cross-validated random forest model, LCO-CB: Leave cluster out cross-validated cubist, when models was ran after excluding DEM.

---

## Author Comment (AC2) · 21 Feb 2020

Dear Dr. Stamm,

Thank you so much for the remarks and below you can find Authors' reply to each comment.

1: Editor comment:

L. 83: The year of the citation of Wong is inconsistent between the main text and the reference list.

Authors' reply:

The correct citation year is 2015; however, the reference will be removed from the

manuscript, as it was misplaced in line 83.

2: Editor comment:

L. 156 - 157 The number of models seems to be inconsistent.

Authors' reply:

In total, two algorithms were used and six models were trained with three different resampling or cross-validation methods. The correct number is six and line 157 will be corrected.

3: Editor comment:

L. 190 - 191: What could be reasons why KF performs worse? Is there a spatial bias?

Authors' reply:

As indicated on Lines 226, 265 and presented on Table 2, the highest accuracy was achieved with k-fold (KF) cross-validated models. The lines 190 and 191 will be reformulated to better stress that KF yields a higher accuracy.

4: Editor comment: L. 238: What is the basis for this statement?

Authors' reply: Adhikari et al., 2013. The citation is missing in this line and will be added to the revised manuscript.

5: Editor comment:

L. 244: Does the predictive value of DEM-derived indices not depend very much on the spatial support and resolution of the data? Have you calculated these indices as averages across the catchments?

Authors' reply:

In the manuscript, we used the DEM-derived covariates at the point of the drainage outlet. We agree that it would be more useful to use averages for the drainage catchments,

however, information on the extents of the drainage catchments was not consistently available, and we would have had to exclude a large number of stations to use this approach. The reasoning will be also stated on the revised manuscript.

6: Editor comment:

Fig. 1a: The data seem to separate into two clusters. Do the points with high discharge but rather low percolation have something in common that could explain the differences?

Authors' reply:

The figure, which contains the information, is Fig. 2a. Due to the different catchment sizes, the discharge behavior might differ between large and small catchment. For the larger catchments, discharge generated in the pipes might not necessarily flow to the outlet but might re-infiltrate into the soil depending on the spatial variability of the soil in the catchment (e.g. areas that are sandier where the natural drainage capacity (drainage class) of the soil is higher). Some of the drainage stations are draining large catchments, which could explain the clustering (Fig. 2b) when the percolation (Db) is compared to drainage discharge (Q).

7: Editor comment:

Fig. 1a&b: Combining the two data suggests that drainage discharge is well correlated (and predicted) by the amount of precipitation. How does this relationship look like if you additionally distinguish between clay and sandy soils?

Authors' reply:

The Figure, which contains the information, is Fig. 2a&b. Fig. 2a demonstrates the correlation between measured drainage discharge (Q) and calculated percolation (Db), and Fig. 2b shows the correlation between measured precipitation (P) and calculated Db. Here we attach an extra plot where the correlation between measured P and Q is demonstrated (Figure 1). We have included the soil type as a predictor in the model

but an extra figure showing the relation between discharge and clay percent could be included in the revised manuscript. The analysis to distinguish between clay and sandy soils will be carried out and included in the revised manuscript.

8: Editor comment:

Tab. 1: Please provide the distribution of predictors (as supporting information).

Authors' reply:

Mean values for all the covariates, excluding the categorical ones, is inserted in Table 1. Based on the comments from Referee #1, depth of sinks (BS) will be excluded from the covariates.

Kind Regards,

Authors

Please also note the supplement to this comment:
https://www.hydrol-earth-syst-sci-discuss.net/hess-2019-650/hess-2019-650-AC2-supplement.pdf

————————————————

[Figure]

**Fig. 1.** Correlation between measured precipitation (P) and measured drainage discharge (Q)

| Predictors | Description | Range/ Class | Mean |
|---|---|---|---|
| Db | Percolation/Discharge out of the root zone (mm y$^{-1}$) | 0 − 1033 | 336 |
| Geological_R | Geological region | 7 classes | - |
| DEM | Elevation (m) | 0.74 − 83.16 | 30.53 |
| Geological_C | Geology of the area | 10 classes | - |
| F_Accu | Flow Accumulation/Number of unslope cells | 1 − 1108 | 14.48 |
| SagaWI | SAGA Wetness Index | 12.16 - 16.58 | 14.17 |
| TWI | Topographic Wetness Index | 3.47 − 12.33 | 5.43 |
| D_Class | Drainage class | 5 classes | - |
| Clay A %† | Clay content 0-30 cm soil depth | 3 − 20.3 | 13.29 |
| Clay B %† | Clay content 30-60 cm soil depth | 2 − 29.1 | 18.22 |
| Clay C %† | Clay content 60-100 cm soil depth | 1.5 − 31 | 19.5 |
| Clay D %† | Clay content 100-200 cm soil depth | 2.2 − 32.6 | 18.91 |
| DDJD LER-A%‡ | Clay content in A horizon | 3 − 24.8 | 14.34 |
| DDJD LER-B%‡ | Clay content in B horizon | 0 − 31.97 | 18.46 |
| DDJD LER-C%‡ | Clay content in C horizon | 0 − 29.1 | 20.94 |
| JB | Danish soil classification for the A horizon | 12 classes | - |
| Gwd_Int | Depth to groundwater table interpolated from well observations and surface water (m) | 0 − 25.31 | 7.42 |
| Wetlands | 0: Non-wetlands; 1: Wetlands; 2: Central wetlands; 3: Peatlands. | 4 classes | - |
| D_DK_New | Artifical drainage-new map | 2 classes | |
| DP_New | Drainage probability-new map | 0 − 0.86 | 0.72 |
| D_DK | Artifcial drainage-old map | 2 classes | - |
| DP | Drainage probability-old map | 0 − 0.82 | 0.72 |
| Demdetrend | Elevation minus the mean elevation in a 4 km radius (m) | -11.4 − 26.04 | 6.23 |
| Dirinsola | Direct insolation (kWh/year) | 1150.08 − 1348.61 | 1273 |
| Gwd_model | Depth to groundwater from the model (m) | 0 − 32.42 | 5.54 |
| Hdtochn | Horizontal distance to the nearest waterbody (m) | 0 − 1114.89 | 324 |
| Midslppos | Mid-slope position | 0 − 0.7 | 0.25 |
| Mrvbf | Multi-resolution index of valley bottom flatness | 0.07 − 8.68 | 3.69 |
| Slpdeg | Surface slope gradient (degrees) | 0.09 − 7.53 | 1.46 |
| Slptochn | Downhill gradient to the nearest waterbody (m) | 0 − 3.48 | 1.20 |
| Vdtochn | Vertical distance to the nearest waterbody (m) | 0 − 19.28 | 6.11 |
| Valldepth | Valley depth (m) | 2.43 − 21.35 | 4.97 |
| Landscape | Landform types | 11 classes | - |

**Fig. 2.** Table 1. List of covariates used to predict the discharge including a description of the parameter and a range specifying the type of covariate.

---

## Author Comment (AC3) · 21 Feb 2020

We greatly appreciate the time and efforts of the referee to provide us with such a comprehensive and detailed review. Undoubtedly, the review will enable us to improve the manuscript.

In the following, we provide replies to the referee's comments. We also describe the changes, which we will make in the final manuscript to accommodate the referee's comments.

1: Referee comment:

L20: I have to disagree that this work opens up for a better understanding of the dynamics of the drainage discharge. As I discuss later, the Authors use average values

throughout the observation period, and therefore, dynamic effects are loss.

Authors' reply:

We fully agree that the sentence is formulated in a wrong way. As the discharge is annual in different time periods for each station, the study cannot open up for better understanding of the "dynamics". However, we use 414 observation points, which are the total yearly discharge values and not the average. Since the temporal resolution of our data set only is on a yearly level the dynamic of the discharge relates to yearly differences and difference between catchments.

2: Referee comment:

Section 2.2 The Authors report They used data from 53 stations; 18 stations collect data from 2012 to 2016 and 34 between 1971 and 2009. The number of stations does not sum up to 53.

Authors' reply:

We appreciate the precision on details and the number of older stations must be corrected to 35.

3: Referee comment:

Section 2.2 The problems are that the measurements cover different time periods. The model may have been trained using data from 2016 in one location and data from 1971 in another location. There are no information on discharge trends in the stations, which may be available at those locations with long observations. Anyway, yearly values were calculated for each location neglecting possible trends and variance in data measurements. Can the Authors find a methodology to use only stations that are consistent in time? There are no information on data quality, while possible data gaps may exist. If this was the case, how were they filled? May I please ask the Authors to add a reference to indicate where precipitation measurements and evapotranspiration values come from?

Authors' reply:

We agree that having the same time period for all the stations would have been ideal and we assessed different scenarios regarding this issue, however, one of the main objectives of this project was to have national scale predictions that will enable us to extract a drainage map. In order to have at least one station at the main geological regions in Denmark, we decided to use all the historical data from the available stations. Taking into account the recommendation, we would include (at least some examples) of the discharge trends in the long-term running stations on the revised manuscript. All the meteorological data were measured either at the stations or the nearby stations.

4: Referee comment:

Section 2.2 According to the Authors' hydrological model, percolation out of the root zone is calculated as the difference between precipitation and evapotranspiration. Here, there exist some assumptions which have not been stated. For example, is it valid to neglect irrigation from the model? Is it valid to assume that the crop-specific coefficient Kc=1 to calculate the actual evapotranspiration from the potential one?

Authors' reply:

None of the catchments/fields used in this study were irrigated. The most commonly sown crop in Denmark is winter wheat and the calculations of evapotranspiration were made accordingly. The relevant information is missing on the current version of the manuscript and would be included during the revision.

5: Referee comment:

Section 2.2 L113: May I please ask the Authors to report the accuracies of the digital maps in Table 1? In this regard, the Authors comment at L237 that accuracy error of the digital maps may influence their importance as covariates. If the Authors know the accuracies, They could carry out a sensitivity analysis using the available standard deviation as prior information and assess the prediction outcomes.

Authors' reply:

The accuracy of the maps are reported in the references Adhikari et al., 2013 and Møller et al., 2018. The statement on L237 is missing the relevant citation to Adhikari et al., 2013 and will be included in the revised manuscript. Although carrying out a sensitivity analysis and assessing the prediction outcomes could be very interesting, the only map with high prediction error reported by the author (Adhikari et al., 2013) is the clay content map, which makes such analysis less necessary in the current manuscript.

6: Referee comment:

Section 2.3 How did the Authors integrate numerical and categorical variables? What was the approach followed by the Authors to convert categorical variables to numerical ones? May the Authors discuss what are the implications of such integrations with respect to the final predictions?

Authors' reply:

Both Cubist and Random Forest work by recurrently splitting the dataset. In this process, they can use categorical covariates as they are, and we therefore did not convert the categorical variables to numeric variables. This is the standard approach in the use of these two algorithms.

7: Referee comment:

Section 2.3 L156: Can I please ask the Authors to state how They extracted the covariate importance? In general, which software and packages were used to carry out the study? L169: The Authors report the possibility to use methods to determine the most effective parameters, thus opening the opportunity to reduce the number of covariates. Did the Authors try to rerun the machine learning using a subset of covariates?

Authors' reply:

We fully agree that important information about the importance measures of the predictors is missing on the manuscript and will be included after revision. We chose %IncMSE as the measure of variable importance in the RF model. The %IncMSE indicates the increase in the MSE of prediction, drainage discharge in this study, as a result of one variable being permuted. The higher the value of %IncMSE is, the more important the variable is for the regression of the RF model. For the Cubist model, each predictor had a value of the VarImp (%), which is a linear combination of the usage of each variable in the rule conditions and the linear regression models. We used this value to measure the importance of each predictor in the Cubist model. However, the model was not ran with reduced number of covariates as it would not have an effect. We calculated these two measures using the function varImp in R package caret, which we used for training the models.

8: Referee comment:

Section 3.1 Please add a reference to Table 2 where the accuracies of the methods are reported.

Authors' reply:

In L174 the accuracies with the reference to the table 2 are reported.

9: Referee comment:

Section 3.1 L179: The cluster analysis was an interesting approach. However, the clusters were different in size. Was there a relationship between overall accuracy and number/location of the stations excluded from the training set?

Authors' reply: For Random Forest, there was no correlation between the number of observations in the clusters and the accuracy obtained, both for RMSE and R2 ($p > 0.05$). For Cubist, there was no correlation between RMSE and the number of observations ($p > 0.05$), but R2 was correlated to the number of excluded observations ($R = -0.53$, $p < 0.05$). However, this finding is most likely due to the fact that it easier

to obtain a high R2 with a small number of points. We will therefore not include this finding in the final version of the manuscript.

10: Referee comment:

Section 3.2 L193: Can I please ask the Authors to use one term either percolation or discharge out of the root zone, for clarity?

Authors' reply:

Using two terms was an error and we would make sure to only use "percolation".

11: Referee comment:

Section 3.2 L211: Please use a new Section for the Discussion.

Authors' reply:

A new section will be created for the discussion in the revised manuscript.

12: Referee comment:

Section 3.2 L225-L230: This paragraph seems crucial for the understanding of the predictions but it is difficult to follow. The Authors here discuss the implications of having time-series covering different time-periods. Because Their explanation is not clear, it is difficult to be convinced about Their interpretation. L229: The Authors state that the model is not dependent on climatic forcing. However, this is not because the effects of precipitation and evapotranspiration are accounted for, which are climatic variables.

Authors' reply:

Re-formulation of L225-L229: Highest accuracies were achieved by KF cross-validated RF and CB. However, training the model on 90 % of the data increases the possibility of having the same station in the training dataset and in the test dataset. Leave-station-out guarantees that the target station does not also appear in the training dataset,

however, it would still bias the accuracy assessment as it has similarities with neighboring stations. Which is why leave-cluster-out resampling is the least biased when training the model, as it excludes all the stations within 10 km (as a cluster) from the training dataset.

L229-L230: a wrong statement by authors, which will be excluded from the manuscript after revision. The models are still dependant on the hydrological data as the percolation (Db) is calculated based on climatic and constantly measured data.

13: Referee comment:

Section 3.2 L241: How can the Readers know that the areas with high catchment area are the ones with larger Q/Db? The Authors may use Figure 3 to show such relation? Maybe They could add some text to report the area.

Authors' reply:

In L102 and L108-L112 we give the relevant information. If it does not suffice more explanation will be included in the revised manuscript.

14: Referee comment:

Section 3.2 L248: While it is possible that low-elevated areas are the ones with higher Q, it is difficult to think that distance from groundwater table or the depth to sink (does this refer to the depth to tile drain?) are not significant. Have the Authors tried to remove the DEM as covariate and see how the other covariates rank?

Authors' reply:

We appreciate the exactness of the Referee and it helped us to notice that depth to sink (BS), which is actually irrelevant for the drained areas, should be excluded from the covariates. To assess the effect of excluding DEM, the model was run without the covariate and the results are shown in Figures 1 to 6. The accuracy of the models do not show a noticeable change after excluding the DEM as a covariate. Regarding

the most important predictors, horizontal distance to the channel (Hdtochn) and clay content in the D horizon (Clay.D) appear as the second and third most important variables after precipitation. These two covariates also had high importance in the models that used elevation as a covariate. The results mainly show the adaptive behavior of machine learning models. When an important covariate is missing, the algorithms can to some extent use other correlated covariates to act as proxies. For example, valley depth and vertical distance to channel may act as proxies for elevation.

15: Referee comment:

Section 3.2 L258: I have to disagree with the Reviewers statement: "the possible variations between years as well as outputs in relation to future climatic scenarios can be studied". In fact, there is no analysis with regards of time; yet, the Authors are somehow contradicting Themselves as They stated at L229 that the model does not depend on climatic forcing. Therefore, no studies in relation to future climate scenarios can be carried out.

Authors' reply:

As mentioned earlier, we fully agree and a false statement by the authors will be corrected after revision.

16: Referee comment:

Figure 1 It can be more informative. It would be ideal to have an ID that identifies each location and link the spatial information with the corresponding metadata compiled in a large additional table. It shows that areas are quite far from each other and may follow peculiar dynamics. This strengthen my concern that the number of yearly values and covariates may not suffice to highlight the functioning of the system. Showing how the location cluster affect the predictions may support the interpretation of the results.

Authors' reply:

Suggestion will be taken into account for the revision of the manuscript.

17: Referee comment:

Figure 2 I would suggest to first show the relationship between measured variables (i.e., Q as dependent variable and P and ET0 as independent). At a later stage I would show the relationship between measured Q and predicted Q. Finally, I would show the relationship between Q and significant covariates. As of now, Figure 2b is not informative. It shows the relationship between the Discharge out of the root zone (Db) and the Precipitation, which the latter was used to calculate Db. Authors' reply:

We appreciate the exactness of the comment and the changes will be applied accordingly. We attached a figure demonstrating the relationship between P and Q.

18: Referee comment:

Figure 3 It would be more meaningful if additional information were provided to understand for which conditions Q is greater than P (e.g., low-lying areas, etc...).

Authors' reply:

The relevant information will be provided in the revised manuscript.

19: Referee comment:

Figure 4 I like the idea of showing the maps because they report the gradient The Authors could ease the Readers if the locations were indicated in the maps. It might be valuable to create insets and show scatterplots between measured Q with each covariate reported in the map, with an errorbar to indicate the accuracy of the maps at the location.

Authors' reply:

The suggestion will be considered for the revised manuscript.

20: Referee comment:

Figure 5 This figure makes me wonder: why if I do not use 1 station in the training

set (LSO), I have worse accuracy than when I do not use 10% of the stations in the training set (KF). I think the Authors very briefly discussed this at L225. But I would kindly ask Them to further clarify and expand their analysis on this. Was it about the time coverage? The location? The accuracy of the maps? I believe it is possible to disentangle this.

Authors' reply:

Most of the 53 stations have multiple years of measured drainage discharge and unlike percolation, the other covariates are constant with time. Training the model on 90 % of the data (KF) increases the possibility of having the same station in the training dataset and in the test dataset. On the other hand, Leave-station-out (LSO) guarantees that the target station does not also appear in the training dataset, however, it would still bias the accuracy assessment as it has similarities with neighboring stations.

21: Referee comment:

Figure 7 Panels b,d,f show elevation, which seems to have big range-discrete values. Can the Authors please explain?

Authors' reply:

In Figure 7 of the first version of the manuscript, tile drainage discharge shows a U-shaped relationship with elevation. This shows that topography can have an effect on discharge, which Db does not account for. As we used the 1-dimensional EVACROP model to calculate Db, this covariate does not account for topography. The observed pattern is most likely a combination of several effects. Firstly, higher elevations receive more precipitation, which would increase discharge. Secondly, EVACROP does not account for surface flow. Lower elevations are likely to receive additional water from upslope positions, which would increase discharge. Thirdly, lower elevations will often have a shallower depth to the groundwater, and groundwater flow from higher positions may therefore contribute to the increased discharge. Together, these explanations show why intermediate elevations may have less discharge than higher and lower elevations.

22: Referee comment:

Table 2. Is there a p-value? Such value could be used to decide which features are significantly relevant and control possible false discovery rates.

Authors' reply:

Could we kindly ask the referee to please elaborate a bit on this comment and question?

23: Referee comment:

MINOR COMMENTS L57: Can I please ask the Authors to state by how much was the improvement in terms of performance of the machine learning compared to physically based models to predict tile drainage discharge? L73: Can the Authors please summarize the accuracy to the Readers? L156: Note that, the Authors are not using 6 models, but 2 models and validating each with 3 different methods.

Authors' reply:

L57: In the study of Kuzmanovski et al. (2015) the comparison showed overall improvement in the prediction of discharge through sub-surface drainage systems, and partial improvement in the prediction of the surface runoff, in years with intensive rainfall.

L73: The results of the study by Noi et al. (2017), showed that very high accuracy of Ta estimation (R2 > 0.93/0.80/0.89 and RMSE ~1.5/2.0/1.6 âŮęC of Ta-max, Ta-min, and Ta-mean, respectively) could be achieved with a simple combination of four LST data, elevation, and Julian day data using a suitable algorithm. The summary of the accuracies for this study will be also added to the manuscript.

L156: We agree that stating six "models" might cause misunderstandings and we would try to explain better in the revised manuscript. We use two different algorithms

and we train six models with different resampling methods.

Please also note the supplement to this comment:
https://www.hydrol-earth-syst-sci-discuss.net/hess-2019-650/hess-2019-650-AC3-supplement.pdf

[Figure]

**Fig. 1.** Model performance of KF-RF: K-Fold cross-validated random forest model, KF-CB: k-fold cross-validated cubist model, when model was ran after excluding DEM.

[Figure]

**Fig. 2.** Model performance of LSO-RF: Leave station out cross-validated random forest model, LSO-CB: Leave station out cross-validated cubist model, when models was ran after excluding DEM.

[Figure]

**Fig. 3.** Model performance of LCO-RF: Leave cluster out cross-validated random forest model, LCO-CB: Leave cluster out cross-validated cubist model, when models was ran after excluding DEM.

[Figure]

**Fig. 4.** Top 10 most important covariates for KF-RF: K-Fold cross-validated random forest model, KF-CB: k-fold cross-validated cubist model,when model was ran after excluding DEM.

[Figure]

**Fig. 5.** Top 10 most important covariates for LCO-RF: Leave cluster out cross-validated random forest model, LCO-CB: Leave cluster out cross-validated cubist model, when model was ran after excluding DEM.

[Figure]

**Fig. 6.**

---

## Author Comment (AC4) · 21 Feb 2020

In order to fit the plots into the online format of the document, figures are separated. Where we reference to Figure 1 and 2 in the reply, it corresponds to Figures 1 to 6. Where we reference to Figure 3 and 4 in the reply, it corresponds to Figures 7 to 12. We apologize for the error and any possible confusion.

---

## Author Comment (AC5) · 2 Mar 2020

8: Referee comment:

Section 3.1 Please add a reference to Table 2 where the accuracies of the methods are reported.

Authors' reply:

We noticed that the accuracy measures reported on Table 2 and L174 are not consistent and the reference to the table in the text will be corrected accordingly.
* * *